# Study on Fluorescence Recognition of Fe^3+^, Cr_2_O_7_^2−^ and *p*-Nitrophenol by a Cadmium Complex and Related Mechanism

**DOI:** 10.3390/molecules28041848

**Published:** 2023-02-15

**Authors:** Lu Liu, Jian-Min Li, Hui-Jie Wang, Meng-Di Zhang, Yu Xi, Jie Xu, Yuan-Yuan Huang, Bo Zhang, Ying Li, Zhen-Bei Zhang, Zi-Fang Zhao, Cheng-Xing Cui

**Affiliations:** 1School of Chemistry and Chemical Engineering, Henan Institute of Science and Technology, Xinxiang 453003, China; 2School of Resources and Environment, Henan Institute of Science and Technology, Xinxiang 453003, China

**Keywords:** coordination complex, fluorescence recognition, mechanism

## Abstract

The effective detection of environmental pollutants is very important to the sustainable development of human health and the environment. A luminescent Cd(II) coordination complex, {[Cd(dbtdb)(1,2,4-H_3_btc)]·0.5H_2_O}*_n_* (**1**) (dbtdb = 1-(2,3,5,6-tetramethyl-4-((2-(thiazol-4-yl)-2H-benzo[d]imidazol-3(3aH)-yl)methyl)benzyl)-2,7a-dihydro-2-(thiazol-4-yl)-1H-benzo[d]imidazole, 1,2,4-H_3_btc = 1,2,4-benzenetricarboxylic acid), was obtained by hydrothermal reactions. Complex **1** has a chain structure decorated with uncoordinated Lewis basic O and S donors and provides good sensing of Fe^3+^, Cr_2_O_7_^2−^, and *p*-nitrophenol with fluorescence quenching through an energy transfer process. The calculated binding constants were 3.3 × 10^3^ mol^−1^ for Fe^3+^, 2.36 × 10^4^ mol^−1^ for Cr_2_O_7_^2−^, and 9.3 × 10^3^ mol^−1^ for *p*-nitrophenol, respectively. These results show that **1** is a rare multiresponsive sensory material for efficient detection of Fe^3+^, Cr_2_O_7_^2−^, and *p*-nitrophenol.

## 1. Introduction

With the continuous development and progress of human society, human activities have more and more interference and influence on the natural environment, and a large number of pollutants have gone far beyond the self-purification function of the natural environment, resulting in serious environmental pollution problems.

Environmental pollution is threatening human life, and health and has become a major concern of the whole world. The detection and monitoring of these environmental pollutants is of great significance to prevent environmental pollution incidents and ensure social security. There are many types of environmental pollutants, such as metal cations, anions, organic compounds, and so on. Metal ions play a vital role in the development of human society. Metal ions discharged into the environment enter the human body through a variety of ways. An appropriate amount of ions is beneficial to the sustainable development of the environment and human health. In a lot of metal ions, Fe^3+^ is an important metal center in catalysis and biotechnology and plays an important role in living organisms, such as in hemoglobin formation [1,2,3]. Moderate Fe^3+^ intake can protect against certain diseases, such as heart disease and Parkinson’s disease [4]. However, Fe^3+^ excess or deficiency can also lead to various diseases [5,6]. For example, excess Fe^3+^ can lead to health problems such as depression and coma. Furthermore, the release of excess Fe^3+^ in industrial wastewater can cause environmental harm. Therefore, Fe^3+^ detection is important for environmental protection.

Dichromate (Cr_2_O_7_^2−^) is an important organic substance that can be used to make dyes, paints, flexible plastics, and cellulose. They can also be used to make substances such as antimicrobials, antioxidants, and antiviral agents. In addition, dichromate can also be used in the manufacture of anticorrosive agents, antioxidants, and antiheat agents. Although Cr_2_O_7_^2−^ is important in many ways, it is one of the most common and dangerous oxygen-containing anions in industrial effluents and is an environmental non-biodegradable pollutant. Dichromate can cause harm to the human body through different ways, such as invading the human body through the digestive tract, respiratory tract, and skin, causing adverse reactions such as nausea and vomiting, and even causing more serious harm, such as visceral damage and circulatory system diseases, when accumulated in the human body [7]. In addition, it can accumulate in plants and animals through biological enrichment and pose a long-term threat to the ecological environment. Therefore, it is urgent to explore effective detection methods for Cr_2_O_7_^2−^.

Nitro compounds can be used as medicine, dyes, spices, explosives, and other industrial chemical raw materials and reagents in organic synthesis. Aromatic nitrocompounds are the raw materials for preparing aromatic amines and diazo salts. Polynitrocompounds are explosive and can be used as explosives; other polynitroso compounds have strong fragrances and can be used to make artificial musk. Although nitrocompounds are very versatile, they are all toxic to humans. In the production site, such compounds mainly pollute the air in the form of dust or vapor, which is inhaled into the human body through the respiratory tract and causes poisoning, and some pollute the skin with liquid, which is absorbed by the skin and causes poisoning [8,9]. Therefore, the development of an economical and efficient method for nitro-molecules detection is extremely important.

Presently, several conventional methods have been used for the detection of Fe^3+^, Cr_2_O_7_^2−^, and nitro-molecules, including chromatography [10] and electrochemical methods [11]. However, these methods inevitably have various problems, such as complex sample pretreatment, large amounts of organic solvents, easy secondary pollution, time-consuming procedures, nonportable testing instruments, skilled operation, and high cost, which limit their application in routine analysis and detection. Thus, a simple and efficient detection method is needed.

In recent years, fluorescence sensing has provided a powerful means to monitor pollutants in the environment. Some traditional fluorescent materials have disadvantages such as poor solubility and severe photobleaching, which limit their practical application in the detection of pollutants by fluorescence sensing. Metal–organic frameworks (MOFs) refer to the crystal porous materials with periodic network structure formed by the self-assembly of transition metal ions and organic ligands. It has the advantages of high porosity, low density, large surface area, regular pore, adjustable pore size, diversity of topological structure, and clipping ability. Because their rich π-conjugated systems comprise organic ligands and optimized pore sizes and shapes, MOFs show great potential in luminescence sensing. Fluorescence sensing technology based on fluorescent MOFs has advantages of short response times, low cost, high sensitivity, and high efficiency. Owing to these obvious advantages over traditional detection methods, the preparation of new luminescent MOFs materials for the detection of various pollutants is of both scientific significance and practical value.

Organic ligands are extremely important in the synthesis of fluorescent MOFs. Organic ligands bearing aromatic or conjugated π-moieties can endow MOFs with outstanding optical properties, resulting in improved high-efficiency recognition of target analytes [12,13]. Based on the above considerations, multidentate ligand dbtdb (1-(2,3,5,6-tetramethyl-4-((2-(thiazol-4-yl)-2H-benzo[d]imidazol-3(3aH)-yl)methyl)benzyl)-2,7a-dihydro-2-(thiazol-4-yl)-1H-benzo[d]imidazole) was selected as the main ligand for this study (Figure 1). Ligand dbtdb bears a 2-(4-thiazolyl)benzimidazole group with a strong chelation ability, groups that can freely twist around two –CH_2_– moieties with diverse angles to form disparate conformations, and coexisting benzimidazolyl and thiazolyl groups that increase the degree of conjugation, resulting in optimized photophysical properties. As we all know, organic ligands of carboxylic acids always show a variety of coordination patterns, so they have been favored by chemical researchers. 1,2,4-H_3_btc contains three carboxyl groups and is a multidentate O-donor coligand. This ligand can coordinate with metal centers through diversiform bridging/chelating modes, which facilitates the construction of extraordinary structures (Figure 1). In addition, the most important parameters for MOF synthesis are temperature, solvent, matrix composition/concentration, degree of solubility of reactants in solvent, and pH of solution. Although experience often dictates the optimal conditions for growing these crystal frames, experimental methods and trial and error methods are still necessary. Accordingly, by introducing 1,2,4-H_3_btc into the Cd(II)/btbb system and constantly experimenting with various synthesis methods, complex {[Cd(dbtdb)(1,2,4-H_3_btc)]·0.5H_2_O}*_n_* (**1**) was obtained under hydrothermal conditions, with the aim to explore its fluorescence detection of different solvent molecules, metal cations, anions, and organic pollutants. Complex **1** exhibited a 1D chain structure ornamented with uncoordinated Lewis basic O and S donors. The fluorescence sensing behavior of complex **1** was researched in water and EtOH, showing notable sensing capabilities for Fe^3+^, Cr_2_O_7_^2−^, and *p*-nitrophenol with fluorescence quenching. Furthermore, the mechanism of luminescence quenching was discussed.

## 2. Results

### Crystal Structure of {[Cd(dbtdb)(1,2,4-H_3_btc)]·0.5H_2_O}_n_
*(**1**)*

As shown in Figure 1a, the asymmetric unit contained one Cd(II) ion, one dbtdb, one 1,2,4-Hbtc^2−^, and a half lattice H_2_O molecule. Each Cd(II) adopted a distorted octahedral coordination geometry finished by two O atoms (O1, O6) from three 1,2,4-Hbtc^2−^ ligands and four N atoms (N1, N2, N5, N6) from two dbtdb ligands. The Cd–O/N bond lengths were in the range of 2.240(3)–2.419(3) Å, while O/N–Cd–O/N bond angles were in the range of 67.32(12)–161.48(12)°, which coincided with reported Cd(II) complexes [14].

Ligand dbtdb (with an N_donor_···N–C_sp3_···C_sp3_ torsion angle of 152.427° and 143.036°) adopted an asymmetric *cis*-conformation. The two dbtdb ligands showed bidentate binding to link the two imminent Cd(II) ions, affording a 26-membered ring with a Cd···Cd separation of 13.8606 Å (Figure 1b). Ligand 1,2,4-btc^3−^ adopted a *μ_2_-η^1^:η^1^:η^0^* mode. The Cd(II) ions were connected by carboxyl oxygen atoms from the two 1,2,4-Hbtc^2−^ ligands, producing a 16-membered ring with a Cd···Cd separation of 9.3763 Å (Figure 1c). The two differently membered rings were connected by Cd(II) atoms, forming a 1D chain along the a-axis (Figure 1d). The adjacent 1D chains were extended into a 2D layer by π···π stacking interactions with a center-to-center separation of 3.865 Å (dihedral angle, 15.075°) between neighboring phenyl rings from ligand 1,2,4-btc^3−^ (Figure 1e). As shown in Figure 1f, under the influence of weak van der Waals interactions, these 2D layers were stacked into a 3D supramolecule. Furthermore, the π–π interactions in each chain between the phenyl rings of 1,2,4-btc^3−^ ligands make the structure of **1** more stable. In **1**, the O and S donors in ligand 1,2,4-btc^3−^ and dbtdb do not coordinate to the Cd(II) ions. Thus, uncoordinated O and S can act as Lewis bases to recognize various analytes.

## 3. Discussion

### 3.1. PXRD Analysis of Complex ***1***

PXRD analysis was performed to check the purity of **1** (Appendix A). The peak positions of the as-synthesized sample were aligned with those simulated. Crystals of **1** were stable in air and did not dissolve in water or ethanol. To check the chemical stability of **1**, each finely ground powder sample of **1** was immersed in H_2_O and ethanol for 24 h, and then PXRD analysis of each sample was conducted (Appendix A). The unchanged PXRD patterns showed that the crystallinity of **1** was retained after solvent treatment, indicating its high chemical stability.

### 3.2. Thermal Analysis

Thermogravimetric analysis (TGA) was conducted to determine the thermal stability of **1** (Appendix A). The TGA curve of **1** exhibited an initial weight loss of 0.61% from 56 to 186 °C owing to the release of half a lattice water molecule (calculated, 0.51%). The further weight loss from 312 to 735 °C was attributed to structure decomposition. CdO residue corresponding to 13.26% of the initial weight (calculated, 14.42%) was observed.

### 3.3. Photoluminescence Properties

The fluorescence properties of compounds with d^10^ metal centers have attracted great interest [15]. Therefore, photoluminescent spectra of free dbtdb ligand, 1,2,4-H_3_btc ligand, and complex **1** were recorded in an aqueous environment, as shown in Figure 2. Free ligand dbtdb showed an emission band at 381 nm (λ_ex_ = 260 nm), 1,2,4-H_3_btc showed a different emission peak at 386 nm (λ_ex_ = 274 nm), and complex **1** exhibited an emission maximum at 378 nm (λ_ex_ = 275 nm). For **1**, the emission spectrum was blue-shifted (by 3 nm and 8 nm) with respect to free dbtdb and 1,2,4-H_3_btc. This blue shift of the emission peaks of **1** indicated that the coordination of dbtdb and 1,2,4-H_3_btc ligands to Cd(II) ions increased the rigidity, leading to less skeleton vibrations and reduced energy loss through radiationless decay of the intraligand emission excited state [16].

The fluorescent behavior of **1** as a suspension in various organic solvents was studied systematically. Complex **1** (1.5 mg) was finely ground, soaked in an organic solvent (3 mL) such as acetone, and then dispersed, adopting ultrasonication for about 15 min. In different solvents, the fluorescent properties of **1** were monitored under the excitation of 275 nm, and the emission curves are shown in Figure 3. Complex **1** clearly exhibited different fluorescence intensities in different solvents. Acetone/methylbenzene quenched the fluorescence of complex **1**. The decreased or quenched fluorescence intensity of complex **1** was hypothesized to be caused by energy transfer from the organic ligand to acetone or methylbenzene molecules [17].

To further explore the recognition of Fe^3+^, Cr_2_O_7_^2−^, and *p*-nitrophenol by complex **1** in water/ethanol, the fluorescence stability of complex **1** was tested in both water and ethanol (Appendix A). The results show that complex **1** exhibited good stability in water and ethanol.

### 3.4. Sensing of Metal Ions

Owing to its Lewis basic O and S active sites, better solvent stability, and fluorescence, complex **1** is a potential candidate material for fluorescence sensors. First, the application of **1** to metal ion detection was investigated. Powdered **1** (1.5 mg) was placed independently into M^x+^ aqueous solutions (3 mL, 10^−3^ mol L^−1^; M^x+^ = Zn^2+^, Ni^2+^, Mn^2+^, Cr^3+^, Na^+^, Co^2+^, Ba^2+^, Pb^2+^, Sr^2+^, Cd^2+^, Cu^2+^, Ca^2+^, K^+^, Bi^2+^, Mg^2+^, Ag^+^, Hg^2+^, and Fe^3+^), and the luminescence responses toward these diverse metal ions were tested. These metal ions had diverse impacts on the fluorescence intensities of **1** (Figure 4a). The fluorescence was almost completely quenched by Fe^3+^ ions, while the other metal ions have low or moderate quenching effects on emission, showing that **1** can be used as a sensory material with good response for Fe^3+^ ions.

The ability of **1** to sensitively detect Fe^3+^ heartened us to further survey its selectivity for Fe^3+^. Experimentally, equivalent amounts of Fe^3+^ and each of the other previously mentioned metal ions were mixed with **1**. In each case, the fluorescence was overtly quenched (Figure 4b). The results show that other metal ions had some interference except for Sr^2+^, Mn^2+^, K^+^, Bi^2+^, Mg^2+^, Pb^2+^, and Cr^3+^, which manifest that **1** had certain selectivity ability for Fe^3+^.

The sensing sensitivities of **1** for Fe^3+^ was determined using the following experiment. A sample of **1** was dispersed in H_2_O (0.5 mg mL^−1^) and subjected to ultrasonic treatment to obtain a suspension. Different amounts of Fe^3+^ solution (0.1 mol L^−1^) were then added to the above suspension liquid, and emission spectra were gained (Appendix A). With incremental addition of Fe^3+^, the emission intensity gradually decreased. Based on the fitted linear equation I_0_/I = 1.103 + 3.3 × 10^3^ [M], the association constant was calculated to be 1.812 × 10^4^ mol^−1^, where I_0_ and I are the fluorescence strength before and after analyte incorporation, respectively, and [M] is the analyte concentration (Appendix A). The quenching coefficient was calculated to be 3.3 × 10^3^ mol^−1^ for Fe^3+^, while the limit of detection (LOD) for Fe^3+^ was 1.1 × 10^−3^ M.

### 3.5. Sensing of Anions

The effect of anions on the luminescence intensity of **1** was determined using various anions (I^−^, C_2_O_4_^2−^, HSO_3_^−^, Br^−^, SO_4_^2−^, F^−^, NO_2_^−^, H_2_PO_4_^−^, S_2_O_8_^2−^, SCN^−^, Cl^−^, Ac^−^, and Cr_2_O_7_^2−^ were selected). Samples of **1** (1.5 mg) were dispersed in water solutions containing the above anions (3 mL, 10^−3^ M). Cr_2_O_7_^2−^ ions clearly quenched the emission of **1**, while other anions caused small to moderate changes in the fluorescence degree of **1**, showing the potential Cr_2_O_7_^2−^ detection ability of **1** (Figure 5a). Next, the sensing selectivity of Cr_2_O_7_^2−^ was then explored through competition experiments. Upon adding Cr_2_O_7_^2−^ to the mixture of **1** with all of the other aforementioned anions, the emission was quenched greatly, indicating the high selectivity of **1** for Cr_2_O_7_^2−^, even in the existence of other interfering anions (Figure 5b).

In order to further study the sensing capability of **1** for Cr_2_O_7_^2−^, fluorescence titrations were executed by adding different volumes of Cr_2_O_7_^2−^ aqueous solution (0.1 mol L^−1^) to a suspension of **1**. The emission intensity of **1** was gradually quenched by the incremental addition of Cr_2_O_7_^2−^ (Appendix A). The quenching equation was written as I_0_/I = 1.235 + 2.36 × 10^4^ [M] (Appendix A), the quenching coefficient was calculated to be 2.36 × 10^4^ mol^−1^, and the LOD was calculated to be 1.6 × 10^−4^ M.

### 3.6. Sensing of Nitro-Molecules

The effect of organic pollutants on the luminescence intensity of **1** was determined using various organic pollutants (4-aminophenol, 2,4,6-Trichlorophenol, *o*-aminophenol, 2,4-dichlorophenol, phenol, carbaryl, *p*-chlorophenol, *m*-dihydroxybenzene, atrazine, 2,6-dichloro-4-nitroaniline, and *p*-nitrophenol were selected). Samples of **1** (1.5 mg) were dispersed in aqueous solutions, including the above organic pollutants (3 mL, 10^−3^ M). *p*-nitrophenol clearly quenched the emission of **1**, while the other organic pollutants led to small to moderate changes in the fluorescence intensity of **1**, showing the potential *p*-nitrophenol detection ability of **1** (Figure 6a). The sensing selectivity for p-nitrophenol was then explored using competition experiments. Upon adding *p*-nitrophenol to a mixture of **1** and all of the aforementioned organic pollutants, the emission was greatly quenched, showing the high selectivity of **1** for *p*-nitrophenol, even in the presence of other interfering organic pollutants (Figure 6b).

To further investigate the sensing capability of **1** toward *p*-nitrophenol, fluorescence titrations were conducted by adding different volumes of *p*-nitrophenol aqueous solution (0.1 mol L^−1^) to a suspension of **1**. The emission intensity of **1** was gradually quenched with incremental addition of *p*-nitrophenol (Appendix A). The quenching equation was written as I_0_/I = 0.617 + 9.3 × 10^3^ [M] (Appendix A), the quenching coefficient was calculated to be 9.3 × 10^3^ mol^−1^, and the LOD was calculated to be 3.2 × 10^−4^ M.

### 3.7. Sensing Mechanism

The fluorescence recognition and sensing of complexes are mainly based on the detected complexes, identifying the enhancement or attenuation of luminescence intensity, and even quenching of luminescence peak position shift caused by the interaction between species and complexes. The interaction between the complex and detected or recognized species includes adsorption, host–guest interaction, and coordination with the unsaturated metal center. Due to the existence of these interactions, when the excitation light of a certain wavelength is used, the transfer of electrons or charges or energy between the complex and the species to be detected or recognized can result in obvious changes in the luminescence behavior of the complex, so as to realize the fluorescence recognition and sensing of the complex [18,19]. Here, we also discuss the recognition mechanism of complex 1 in detail, which not only provides a theoretical basis for the content of this experiment but also lays a foundation for the development of complexes in this field.

As shown in Figure 7, a model system denoted as **1-Model** was adopted to understand the sensing ability of the current Cd(II) coordination complex. Theoretical investigations of the geometrical and electronical structure were conducted on **1-Model** (Figure 7) and **1-Model-Fe^3+^** using density functional theory (DFT). The optimized geometry of **1-Model** suggested that the Cd–O bond lengths were 2.460 and 2.256 Å, which is in accordance with the experimental results. The detection of cation, anion, and organic pollutants was based on their interaction with **1**. Eight benzene units were coordinated by Cd(II) in **1-Model**, while **Fe^3+^** was located in the cavity formed by six benzene units in **1-Model-Fe^3+^**. Consequently, the interactions mainly arose from cation–π interactions. The cation–pi interaction is a stabilizing electrostatic interaction of a cation with the polarizable π electronic cloud of an aromatic ring. Cation–π interactions usually have energetically importance when the ligand has aromatic rings and are involved in control of the interaction between ions and the sensor [20]. The lowest unoccupied orbital (LUMO) level was related to anion–π interactions, and the highest occupied orbital (HOMO) level was related to cation–π interactions. The calculated HOMO and LUMO levels of **1-Model** were −5.55 and −1.04 eV, respectively, which corresponds to a HOMO–LUMO gap of 4.51 eV. The HOMO-LUMO gap of **1-Model-Fe^3+^** is 4.60 eV. The size of HOMO–LUMO gap can be used to predict the strength and stability of transition metal complexes, as well as the colors they produce in solution [21]. More importantly, the larger a compound’s HOMO–LUMO gap, the more stable the compound. Consequently, the system becomes more stable when the Fe^3+^ cation is adsorbed on **1-Model**. This is possibly the reason that 1-Model could act as a sensor for Fe^3+^. In particular, the HOMO distribution changed to the vicinity of **Fe^3+^**. The attraction of aromatic surfaces to ions could also be associated with the quadrupole moment perpendicular to the aromatic plane (*Q*_zz_) [22]. The dipole moments of **1-Model** and **1-Model-Fe^3+^** were nearly identical (27.3 Debye). However, the *Q*_zz_ value increased from 46.0 to 61.0 Debye–Angstrom after adsorption of **Fe^3+^** to form **1-Model-Fe^3+^**, indicating that the phenyl ring in **1** acts as a important role in the interaction with Fe^3+^. As complex **1** contains many such adsorption sites, a quasi-self-stabilization effect could continuously occur during the sensing process. The detection of Cr_2_O_7_^2−^ and p-nitrophenol by 1-Model should have a similar mechanism to Fe^3+^. In one word, the ion–π interaction, the HOMO–LUMO gap, and the *Q*_zz_ are important factors for the ability of **1** to detect ions in the current study.

## 4. Materials and Methods

### 4.1. General Information and Materials

All reagents and solvents were commercially available, except for dbtdb, which was synthesized according to the literature [23]. In the region of 400–4000 cm^−1^, FT-IR spectra were tested on an FTIR-7600 spectrophotometer. The content of C, H, and N was recorded on a FLASH EA 1112 elemental analyzer. PXRD were conducted using Cu Kα_1_ radiation on a D8 Advance A25 diffractometer. Thermal analysis was performed on an STA 449 F5 Jupiter thermal analyzer. The fluorescence properties were studied using a Cary Eclipse fluorescence spectrophotometer.

### 4.2. Synthesis of {[Cd(dbtdb)(1,2,4-H_3_btc)]·0.5H_2_O}_n_
*(**1**)*

Cd(NO_3_)_2_·4H_2_O (0.1mmol), dbtdb (0.1 mmol), 1,2,4-H_3_btc (0.1 mmol), and H_2_O (10 mL) were mixed and heated in a 25 mL steel vessel at 160 °C for 3 days. After cooling the mixture, yellowish crystals were obtained in 55% yield (based on Cd). Anal. calcd. for C_82_H_66_Cd_2_N_12_O_13_S_4_ (%): C, 55.31; H, 3.74; N, 9.44; and S, 7.20. Found: C, 54.84; H, 3.56; N, 9.52; and S, 7.04. IR (cm^−1^, KBr): 3055(s), 2906(w), 1697(s), 1583(s), 1544(w), 1473(m), 1440(vs), 1375(w), 1334(s), 1305(w), 1170(w), 1070(w), 1014(m), 833(s), 746(vs), 661(w), 530(w), and 493(w).

### 4.3. Fluorescence Sensing Experiments

Powdered **1** (1.5 mg) was placed into separate aqueous solutions (3 mL) containing various analytes (10^−3^ M). Under the excitation of 275 nm, the photoluminescence responses were recorded. Fluorescence titration was conducted by adding analytes to a suspension of **1** (3 mL).

### 4.4. X-ray Crystallography

Crystallographic data for **1** were collected using an Xcalibur Eos Gemini CCD diffractometer (Cu-Kα, λ = 1.54184 Å) at a temperature of 20 ± 1 °C. Absorption corrections were applied by using the multiscan program. The data were corrected for Lorentz and polarization effects. Structures were solved by direct methods and refined with a full-matrix least-squares technique based on *F*^2^ using the ShelXL crystallographic software package [24]. Then, the coordinates of all nonhydrogen atoms are determined and the anisotropic thermal parameter method is used for refinement by using the total-matrix least-squares method and difference Fourier function method. Then, the position of the hydrogen atom is found by theoretical hydrogenation method, and the hydrogen atom is refined by the knight model. Crystallographic crystal data and structure-processing parameters for **1** are summarized in Appendix A, while selected bond lengths and bond angles for **1** are listed in Appendix A.

### 4.5. Computational Methods

The geometrical structures of **1-Model** and **1-Model-Fe^3+^** were optimized at the theoretical level of cam-B3LYP/6-31G(d) with Gaussian 16 software [25]. The frequency analysis was further performed to confirm that each structure is real local minimum at the same level as optimization. The distribution of HOMO and LUMO were obtained with Chemcraft software (version 1.8).

## 5. Conclusions

A Cd(II) complex with a 1D chain structure was prepared under hydrothermal conditions by a mixed-ligands strategy. The good stability in H_2_O and ethanol, good fluorescence properties, and structure containing uncoordinated Lewis basic O and S atoms of **1** makes it an unusual multiresponsive fluorescence sensor for the detection of Fe^3+^, Cr_2_O_7_^2−^, and *p*-nitrophenol with fluorescence quenching. This research also indicates that the functional atom in a ligand can endow the complex with unique properties.

## Figures and Tables

**Scheme 1 molecules-28-01848-sch001:**
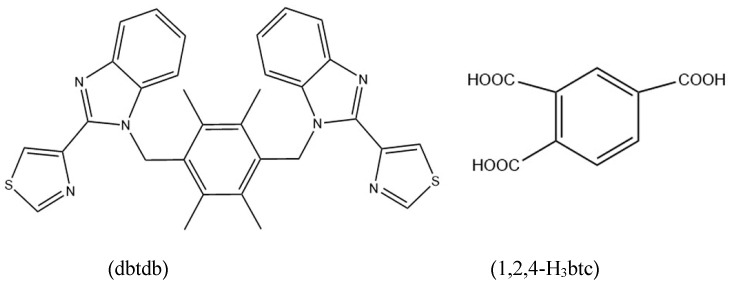
Ligands dbtdb and 1,2,4-H_3_btc used in this study.

**Figure 1 molecules-28-01848-f001:**
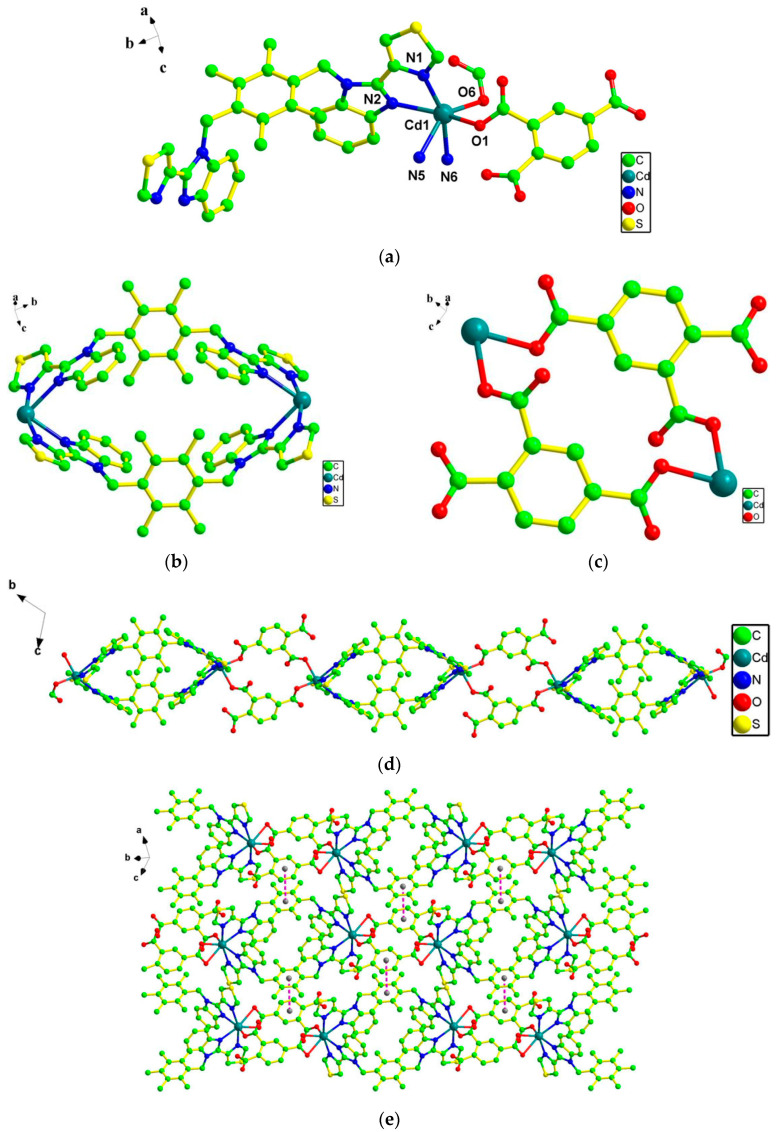
(**a**) Coordination environment diagram around the Cd(II) center in **1**. (**b**) The 26-membered rings constructed by two dbtdb ligands and two Cd atoms. (**c**) The 16-membered rings constructed by two 1,2,4-H_3_btc ligands and two Cd atoms. (**d**) Chain structure of **1**. (**e**) 2D layers of **1**. (**f**) 3D supramolecular structure of **1**.

**Figure 2 molecules-28-01848-f002:**
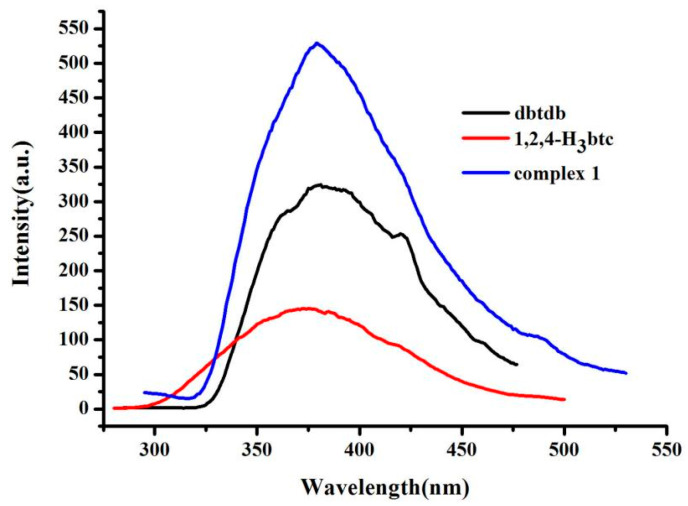
Photoluminescent spectra of free dbtdb, 1,2,4-H_3_btc, and complex **1**.

**Figure 3 molecules-28-01848-f003:**
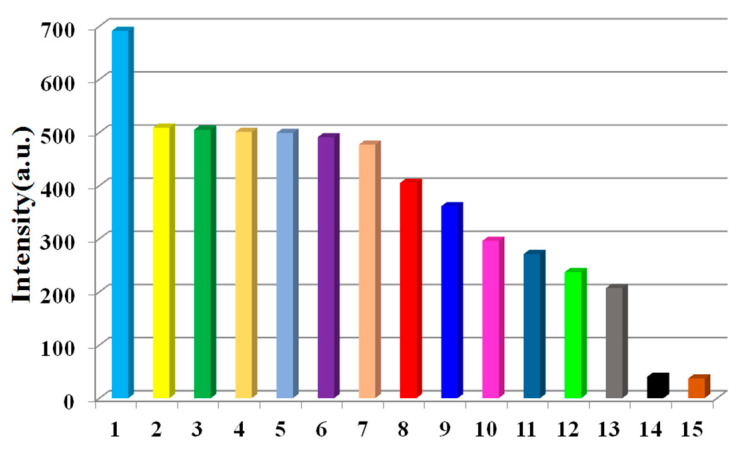
Fluorescence intensity map of complex **1** (1.5 mg) dispersed in aqueous solutions of diverse organic solvents (3 mL) excited at 275 nm: (1) methanol; (2) cyclohexane; (3) acetonitrile; (4) ethanol; (5) 1,4-dioxane; (6) ethyl acetate; (7) water; (8) dichloromethane; (9) trichloromethane; (10) ethylene glycol; (11) DMF; (12) tetrahydrofuran; (13) dimethyl sulfoxide; (14) methylbenzene; and (15) acetone.

**Figure 4 molecules-28-01848-f004:**
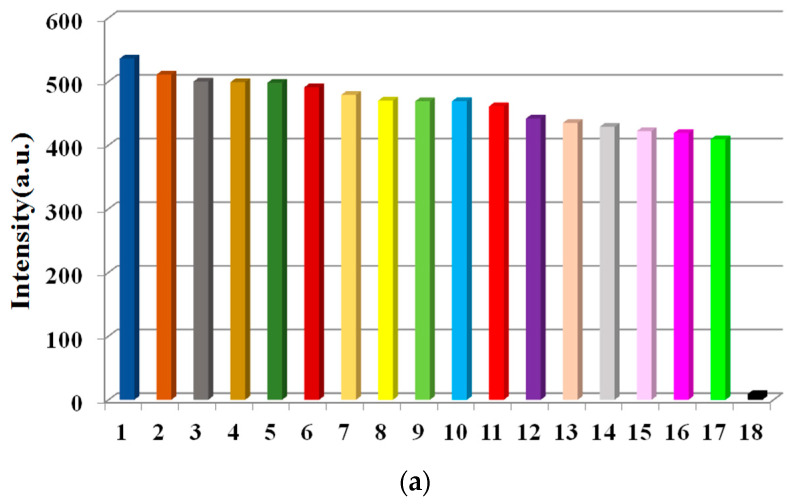
Fluorescence intensity map of (**a**) complex **1** (1.5 mg) dispersed in aqueous solutions of multifarious metal ions (3 mL, 10^−3^ M) under the excitation of 275 nm and (**b**) with subsequent addition of Fe^3+^. (1) Zn^2+^, (2) Ni^2+^, (3) Mn^2+^, (4) Cr^3+^, (5) Na^+^, (6) Co^2+^, (7) Ba^2+^, (8) Pb^2+^, (9) Sr^2+^, (10) Cd^2+^, (11) Cu^2+^, (12) Ca^2+^, (13) K^+^, (14) Bi^2+^, (15) Mg^2+^, (16) Ag^+^, (17) Hg^2+^, and (18) Fe^3+^.

**Figure 5 molecules-28-01848-f005:**
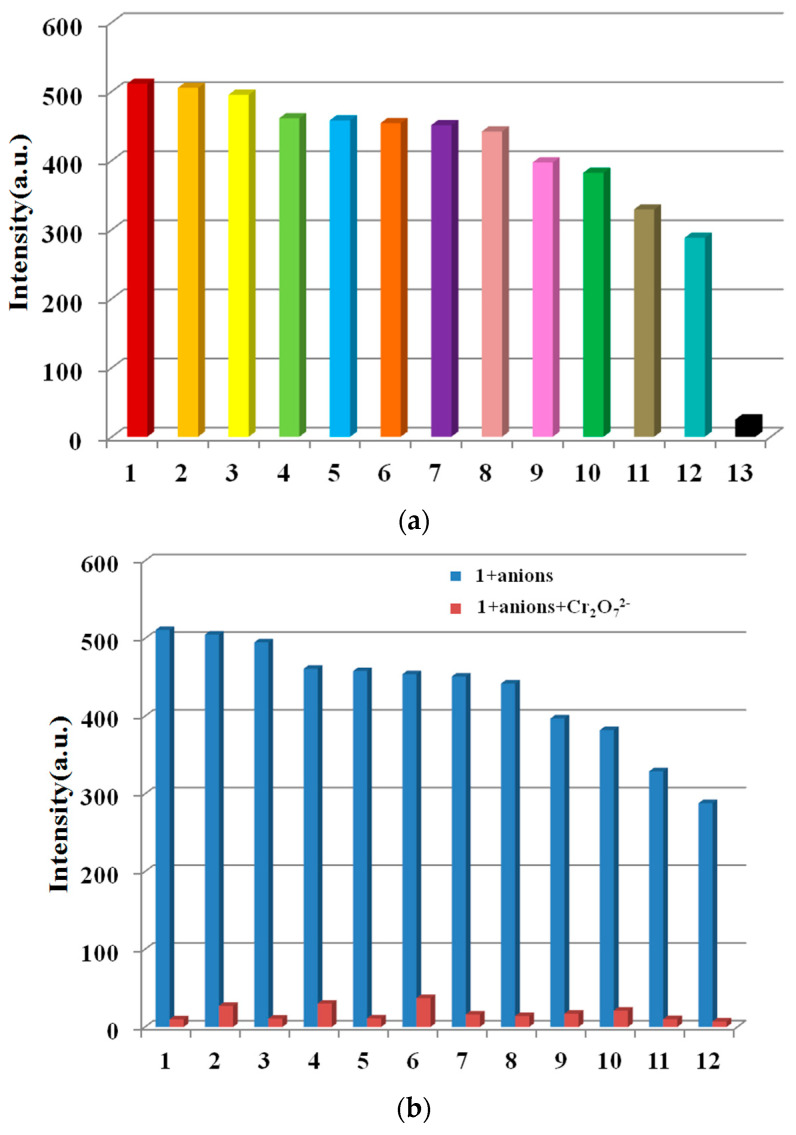
Luminescence intensity map of (**a**) complex **1** (1.5 mg) dispersed in aqueous solutions of multifarious metal ions (3 mL, 10^−3^ M) under the excitation of 275 nm and (**b**) with subsequent addition of Cr_2_O_7_^2−^. (1) I^−^, (2) C_2_O_4_^2−^, (3) HSO_3_^−^, (4) Br^−^, (5) SO_4_^2−^, (6) F^−^, (7) NO_2_^−^, (8) H_2_PO_4_^−^, (9) S_2_O_8_^2−^, (10) SCN^−^, (11) Cl^−^, (12) Ac^−^, and (13) Cr_2_O_7_^2−^.

**Figure 6 molecules-28-01848-f006:**
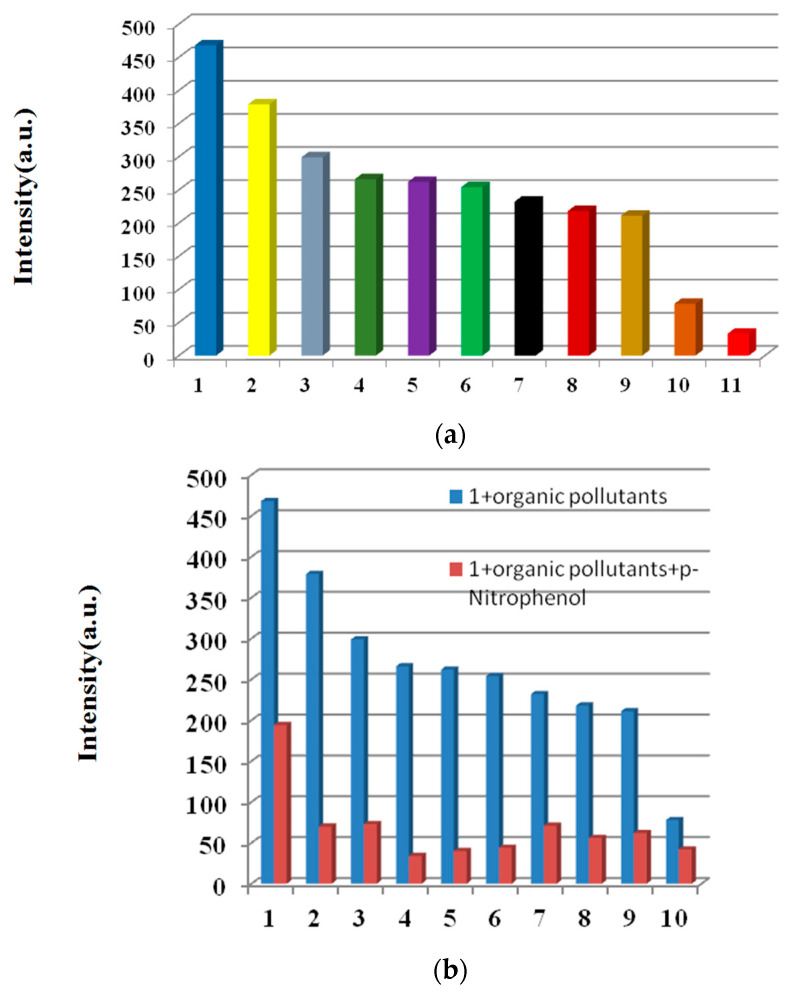
Luminescence intensity map of (**a**) complex **1** (1.5 mg) dispersed in aqueous solutions of multifarious organic pollutants (3 mL, 10^−3^ M) under the excitation of 275 nm and (**b**) with subsequent addition of *p*-nitrophenol. (1) 4-aminophenol, (2) 2,4,6-Trichlorophenol, (3) *o*-aminophenol, (4) 2,4-dichlorophenol, (5) phenol, (6) carbaryl, (7) *p*-chlorophenol, (8) *m*-dihydroxybenzene, (9) atrazine, (10) 2,6-dichloro-4-nitroaniline, and (11) *p*-nitrophenol.

**Figure 7 molecules-28-01848-f007:**
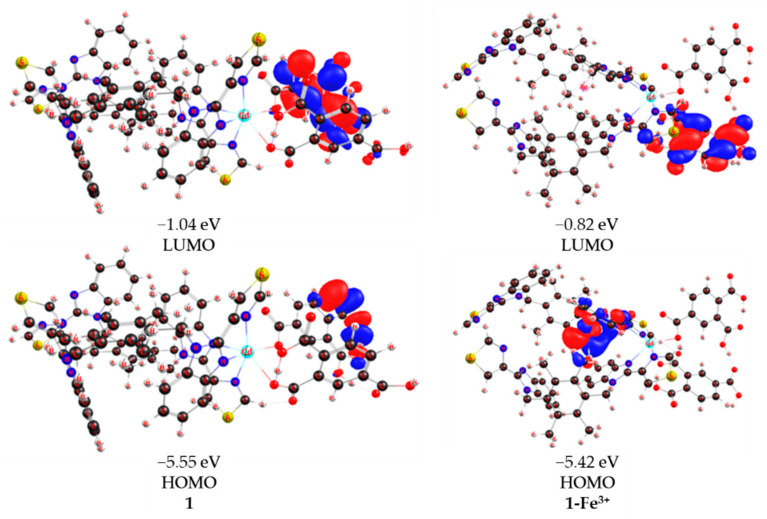
The geometrical structures and the energies and distributions of HOMO and LUMO for **1-Model** and **1-Model-Fe^3+^** were calculated with cam−B3LYP functional.

## Data Availability

Crystallographic data for **1** have been deposited at the Cambridge Crystallographic Data Centre with CCDC reference number 2159587.

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
