# Peer review of "Study on Fluorescence Recognition of Fe^3+^, Cr_2_O_7_^2−^ and *p*-Nitrophenol by a Cadmium Complex and Related Mechanism"

_molecules, 2023, doi:10.3390/molecules28041848_

Round 1

Reviewer 1 Report

Please find the attachment

Author Response

Cheng-Xing Cui

School of Chemistry and Chemical Engineering

Henan Institute of Science and Technology

Xinxiang, Henan, 453003

  1. R. China

Tel: 18625905473

E-mail:chengxingcui@hist.edu.cn

Jan. 19, 2023

Dear Editor :

Thank you very much for your letter and the reviewers’ comments about our manuscript “Study on fluorescence recognition of Fe3+, Cr2O72- and p-nitrophenol by a cadmium complex and related mechanism” (molecules-2149372). We have made a revision of the manuscript according to the comments. We submit here the revised manuscript and our responses to the reviewers’ comments.

Thanks for your time and your kind consideration to this matter.

Sincerely yours,

Cheng-Xing Cui

Response to Referee 1

Thank you very much for your review of our manuscript. We have noted your suggestions and revised the manuscript according to your comments:

  1. In Figs. 4, 5, and 6 the blank signals (with no analyte) are not shown. They are needed to figure out the intensity of the analytical signal.

Response: Thank you for your comments, we again read the paper. In Figure 3, we have given the fluorescence intensity map of complex 1 dispersed in aqueous solutions of diverse organic solvents. Thus, we didn't give the signal repeatedly in Figure 4, 5, and 6.

  1. line 155: little interference is claimed but it contradicts Fig 4b, because the signals with Fe(3+) differ several-fold. For example, the effects of Mn and Ca. This cannot be deemed “little interference”.

Response: Thank you for your comments, we analyzed the results of the experiment carefully. In line 160 and 161, the sentence “The results showed that other metal ions had little interference, which manifest that 1 had superior selectivity for Fe3+.” has been revised to the sentence “The results showed that other metal ions had some interference except for Sr2+, Mn2+, K+, Bi2+, Mg2+, Pb2+ and Cr3+, which manifest that 1 had certain selectivity ability for Fe3+.”. The changes are marked yellow in the text.

  1. Another question, are these effects reproducible? Were any replicated measurements taken? No statistical data are given throughout the text.

Response: Thank you for your comments. At present, there are many articles on fluorescence recognition experiments, and we also refer to relevant references. Although we did not deliberately repeat the experiment alone, during the experiment, we first predicted on the fluorometer to see if the complex could recognize a certain pollutant, and then carried out the full experiment. Therefore, the whole experiment is completely reliable.

  1. The analytical part of work ends up with the reported LODs. These LODs are very high for a fluorimetric method, although the determination may be selective for Fe and Cr(VI). It is necessary to discuss which real-world samples can be analyzed with such limits of detection and selectivity.

Response: Thank you for your comments. At present, our experiment is still in the laboratory stage and has not considered the analysis of pollutants in the real world, such as the identification of pollutants in river water, sewage, food, etc. With the development of this experiment, we will also consider using complex to detect pollutants in river water, sewage and food in the future.

  1. What conclusions can be drawn from Fig. 7?

Response: Thank you for your comments. The interaction between 1 and the ions is crucial to the sensing of the ions by 1. In section 3.5, we theoretically investigated the electronic structure of 1-Model before and after adsorption of Fe3+ with reliable theoretical methods. The calculated HOMO-LUMO gap is 4.51 eV for 1-Model and 4.6 eV for 1-Model-Fe3+. The wider HOMO-LUMO gap of indicates that the more difficult of electron excitation in 1-Model-Fe3+ than in 1-Model. This indicates a large stabilization of Fe3+ to the total system because there are innumerous such adsorption sites in 1. Consequently, the conclusions that can be drawn from Fig. 7 is as followings:

  • The adsorption of Fe3+makes the distribution of HOMO change from the ligand to the metal center.
  • The larger HOMO-LUMO gap indicates that the adsorption of Fe3+on 1 could stabilize the system, which we called a quasi-self-stabilization effect.

To make our discussion more reasonable, we added two sentences to discussion the crucial role of interaction between 1 and ions in the sensing. We highlighted them with yellow color in the revised manuscript.

  1. section 3.5: The whole section is written quite messy and the conclusions on the sensing mechanism are not formulated. Actually, the mechanism is not suggested here.

Response: Thank you for your comments. The theoretical methods are important for the understand of chemical phenomenon and have become a routine technique in the research of chemistry problems. Herein, we combined theoretical and experimental methods to understand the sensing mechanisms. We tried to adopt the ion-π interaction and theoretically calculated electronic structure to discussion the stabilization of Fe3+ to 1-Model. Maybe the discussion was not well organized. Consequently, we carefully revised this part of manuscript and labelled the revised text in yellow.

  1. line 210: Why such a set of organic compunds was selected for the investigation, solving what kind of task was it targeted? No other nitro-compounds were studied; what is the effect of other nitrogroup-containing compounds that are also supposed to act as quenchers?

Response: Thank you for your comments. We chose these organic compounds to study because they're all we can find in the lab right now. We have just started to explore the experiment of fluorescence recognition, and various problems will inevitably occur in this process. In the future, we will improve our experimental scheme according to your suggestions. In addition, nitrocompounds are electron-deficient groups, and complexes can give electrons to nitrocompounds, which can quench the fluorescence of nitrocompounds.

  1. line 218: “Cd–O bond lengths were 2.460 and 2.256 Å, which is in accordance with the experimental results.” – what kind of experimental results are implied here?

Response: Thank you for your comments. In line 224, Cd–O bond lengths are the result of theoretical calculation. In here, the experimental results are the actual Cd–O bond lengths of the complex 1 in line 90.

  1. lines 220-225: cation–πinteractions should be substantiated more clearly.

Response: Thank you for your comments. We have revised the discussion of cation–π interactions, which we labelled in yellow color.

  1. line 3 (title): ‘complexe’ (complex)

Response: Thank you for your comments. In line 3, the word “complexe” has been revised to “complex”. The changes are marked yellow in the text.

  1. The same data are presented in Figures ‘a’ and ‘b’ of Figs. 4, 5, and 6. Figures ‘a’ can be safely omitted.

Response: Thank you for your comments. Although there are repeated parts in Figure a and b of Figure 4, 5, and 6, Figure b allows us to clearly see the experimental result after adding Fe3+, Cr2O72- and p-nitrophenol, and make an obvious comparison.

  1. Lines 140 and 145, line 181: why are cations and anions arranged in such sequences? Logically, they should have been ordered by atomic weight, or by signal intensity, or starting from alkalis and alkaline earths and then moving on to transition metals. But here they are given without any regularity.

Response: Thank you for your comments. When we drew the picture, we didn't put these cations and anions in a certain order. Although these cations and anions do not follow a certain order, but this does not affect our judgment of the results. In the future work, we will draw more beautiful pictures based on your suggestions.

  1. line 158: suffered–? (subjected to)

Response: Thank you for your comments. In line 164, the word “suffered” has been revised to “subjected”. The changes are marked yellow in the text.

  1. line 166, 188: the LODs should be shown with 1, at maximum 2 significant digits (1.1·10–3mol/L).

Response: Thank you for your comments. According to your comments, in line 172, “1.1127´10-3” has been revised to “1.1´10-3”. In line 194, “1.555´10-4” has been revised to “1.6´10-4”. In line 213, “3.226´10-4” has been revised to “3.2´10-4”. The changes are marked yellow in the text.

-------------------------------------------------------------------------------------------------------

Finally, thank you very much for your earnest review of my manuscript. We have learned much knowledge from your comments and advice.

Reviewer 2 Report

This manuscript reports a luminescent Cd(II) coordination complex, which shows fluorescence sensing of Fe3+, Cr2O72-, and p-nitrophenol based on a fluorescence quenching process. I would like to see it publish in this journal after a major revision.

1.      In the introduction, they mention “Metal–organic frameworks (MOFs) are new crystalline materials with 1D/2D/3D net-work structures”, which is not right. The authors should better understand the definition of MOF.

2.      The two free ligands all exhibit blue emission at the solid state, and the author should identify which ligand contribute most to the fluorescence of the as-made compound, whether by DFT calculation or other experimental measurements.

3.      Please provide the methods to get the quenching coefficient and LOD values. As we know, Stern–Volmer (SV) equation (I0/I =1+ Ksv[M]) is usually used in MOF sensing. What is the quenching coefficient in this article, Ksv value? Why the authors adopted an uncommon used equation instead of SV equation in this work? Please give an explanation.

4.      The authors should use SV equation to provide the Ksv value and LOD values, and make a comparative study with other MOF sensors with sensing performances to Fe3+, Cr2O72-, and p-nitrophenol. And I think some tables are needed to summarize the reported results for comparative study.

5.      The sensing experiments for the acid H2PO4and HSO3 anions have also been performed. Please provide the PXRD patterns to check the stability of the compound after sensing of these two acid anions.

6.      Actually, as for sensing p-nitrophenol, it did not show very good selectivity. The compound shows the similar fluorescence quenching effect to 2,6-dichloro-4-nitroaniline in this experimental section. So the nitro group with strong electron withdrawing ability is the matter. Please check how the sensing performance of this compound to other lab used nitro-bearing molecules. If the presented nitro bearing compounds exhibit similar quenching effect, the heading “3.3 Sensing of p-nitrophenol” would be better changed into “nitro-molecules sensing”. And the related third paragraph in introduction part should be revised accordingly.

7.      The sensing mechanism for Cr2O72- and nitro-bearing compounds should be provided.

8.      In Figures S4, S5 and S6, it is better to use the concentration to express the content of additives rather than the volume; Please separate the PXRD patterns in Figure S1, which is not clear to distinguish at the current form.

9.      Other minor issules: 

Figure 1 is not very clear, especially the coordinate axis in a, b, c.

In Figure S3, Chinese characters should revised to English.

Please check table S1, the format for some crystallographic symbol should be monitored.

In the References section, please note formatting issues, such as 7 and 20.

Author Response

Cheng-Xing Cui

School of Chemistry and Chemical Engineering

Henan Institute of Science and Technology

Xinxiang, Henan, 453003

  1. R. China

Tel: 18625905473

E-mail:chengxingcui@hist.edu.cn

Jan. 19, 2023

Dear Editor :

Thank you very much for your letter and the reviewers’ comments about our manuscript “Study on fluorescence recognition of Fe3+, Cr2O72- and p-nitrophenol by a cadmium complex and related mechanism” (molecules-2149372). We have made a revision of the manuscript according to the comments. We submit here the revised manuscript and our responses to the reviewers’ comments.

Thanks for your time and your kind consideration to this matter.

Sincerely yours,

Cheng-Xing Cui

Response to Referee 2

Thank you for your review of our manuscript. We have revised the manuscript according to your comments:

  1. In the introduction, they mention “Metal–organic frameworks (MOFs) are new crystalline materials with 1D/2D/3D net-work structures”, which is not right. The authors should better understand the definition of MOF.

Response: Thank you for your comments. According to your comments, in line 49, the sentence “Metal–organic frameworks (MOFs) are new crystalline materials with 1D/2D/3D network structures.” has been revised to the sentence “Metal-organic frameworks (MOFs) refer to the crystal porous materials with periodic network structure formed by self-assembly of transition metal ions and organic ligands. It has the advantages of high porosity, low density, large surface area, regular pore, adjustable pore size, diversity of topological structure and clipping ability.”. The changes are marked yellow in the text.

  1. The two free ligands all exhibit blue emission at the solid state, and the author should identify which ligand contribute most to the fluorescence of the as-made compound, whether by DFT calculation or other experimental measurements.

Response: Thank you for your comments. From Fig. 7, it could be found that the distributions of LUMOs for 1-Model and 1-Model-Fe3+ are mainly on the ligands that contains phenyl ring. Consequently, the phenyl ring maybe crucial for the emission of blue light at the solid state. However, the precise prediction could only be obtained by Time Dependent Density Function Theory. However, it is very time consuming for such large system herein. Consequently, we added one tentative sentence to answer this question.

  1. Please provide the methods to get the quenching coefficient and LOD values. As we know, Stern–Volmer (SV) equation (I0/I =1+ Ksv[M]) is usually used in MOF sensing. What is the quenching coefficient in this article, Ksv value? Why the authors adopted an uncommon used equation instead of SV equation in this work? Please give an explanation.

Response: Thank you for your comments. We checked the formula in the text carefully. I'm sorry we made such a mistake. Stern–Volmer (SV) equation (I0/I =1+ Ksv[M]) is usually used in MOF sensing. That's the equation we're using in the text. We find that only on line 168, by mistake, we wrote this formula wrong. In line 168, “ I/I0 ” has been revised to “I0/I”. In addition, in Knowledge Information, we also corrected the same error. The changes are marked yellow in the text. The Ksv value is the quenching coefficient in this article.

  1. The authors should use SVequation to provide the Ksv value and LOD values, and make a comparative study with other MOF sensors with sensing performances to Fe3+, Cr2O72-, and p-nitrophenol. And I think some tables are needed to summarize the reported results for comparative study.

Response: Thank you for your comments. Before and after the experiment, we also consulted many relevant literatures, but our experimental results were not outstanding. Therefore, we did not list the identification results of Fe3+, Cr2O72-, and p-nitrophenol from other research groups and compared them with ours. In the future, if we synthesize a complex that has a better recognition effect on pollutants, we will list a table to compare and show our advantages.

  1. The sensing experiments for the acid H2PO4and HSO3 anions have also been performed. Please provide the PXRD patterns to check the stability of the compound after sensing of these two acid anions.

Response: Thank you for your comments. In the “3.1. Photoluminescence properties” section, the stability of the complex in water and ethanol was tested. However, stability in other cations, anions and other contaminants has not been tested. Compared with traditional detection methods, fluorescence sensing technology has the advantages of short response time, low cost, high sensitivity and high efficiency. It is because of the short response time that pollutants have little influence on the stability of the complex in a short time. If the complex is destroyed, then we can not measure the fluorescence intensity of the complex. Therefore, we did not consider the stability of the complex after testing fluorescence recognition experiments. In future experiments, we will consider your suggestions into our experimental scheme.

  1. Actually, as for sensing p-nitrophenol, it did not show very good selectivity. The compound shows the similar fluorescence quenching effect to 2,6-dichloro-4-nitroaniline in this experimental section. So the nitro group with strong electron withdrawing ability is the matter. Please check how the sensing performance of this compound to other lab used nitro-bearing molecules. If the presented nitro bearing compounds exhibit similar quenching effect, the heading “3.3 Sensing of p-nitrophenol” would be better changed into “nitro-molecules sensing”. And the related third paragraph in introduction part should be revised accordingly.

Response: Thank you for your comments. Through literature review, we found that all complexes have fluorescence quenching effect on nitro compounds, but the quenching degree is different. According to your comments, the heading “3.4 Sensing of p-nitrophenol” has been revised to “Sensing of nitro-molecules”. In introduction part, “the third paragraph” has been revised to “Nitro compounds can be used as a medicine, dyes, spices, explosives and other industrial chemical raw materials and reagents in organic synthesis. Aromatic nitrocompounds are the raw materials for preparing aromatic amines and diazo salts. Polynitrocompounds are explosive and can be used as explosives; Other polynitroso compounds have strong fragrances and can be used to make artificial musk. Although nitrocompounds are very versatile, they are all toxic to humans. In the production site, such compounds mainly pollute the air in the form of dust or vapor, which is inhaled into the human body through the respiratory tract and causes poisoning, and some pollute the skin with liquid, which is absorbed by the skin and causes poisoning [8,9]. Therefore, the development of an economical and efficient method for nitro-molecules detection is extremely important.”. The changes are marked yellow in the text.

  1. The sensing mechanism for Cr2O72-and nitro-bearing compounds should be provided.

Response: Thank you for your comments. We tried to theoretically investigate the sensing mechanisms for Cr2O72- and nitro-bearing compounds. However, the computational resources needed is very large and we ran the calculating job about two months on the workstation in our lab but fail because the wave function could not be converged. It encountered a similar situation for the nitro-bearing compounds. We could predict from our experience that the adsorption of Cr2O72- and nitro-bearing compounds could also give a similar quasi-self-stabilization effect to that of Fe3+. Consequently, we added such point of views in the main text to discuss the sensing mechanism for Cr2O72- and nitro-bearing compounds and highlighted them in yellow.

  1. In Figures S4, S5 and S6, it is better to use the concentration to express the content of additives rather than the volume; Please separate the PXRD patterns in Figure S1, which is not clear to distinguish at the current form.

Response: Thank you for your comments. In Figures S4, S5 and S6, the abscissa is something that we drew after a lot of reference. In Figure S1, the PXRD patterns have been separated. The changes are marked yellow in the text.

  1. Figure 1 is not very clear, especially the coordinate axis in a, b, c.

Response: Thank you for your comments. We have revised Figure 1.

  1. In Figure S3, Chinese characters should revised to English.

Response: Thank you for your comments. We have revised the drawing statement of Figure S3.

  1. Please check table S1, the format for some crystallographic symbol should be monitored.

Response: Thank you for your comments. We have checked table S1.

  1. In the References section, please note formatting issues, such as 7 and 20.

Response: Thank you for your comments. We have checked the references.

-------------------------------------------------------------------------------------------------------

Finally, thank you very much for your earnest review of my manuscript. We have learned much knowledge from your comments and advice.

Round 2

Reviewer 1 Report

The authors failed to satisfactorily reply to queries No. 1, 3, 4, 5 (I did not find the changes in text), 6 (not so much has been changed), 7, 11, and 12.

The authors issued such expressions as: “We chose these organic compounds to study because they're all we can find in the lab right now”; “In the future work, we will draw more beautiful pictures based on your suggestions”; “Although we did not deliberately repeat the experiment alone, during the experiment, we first predicted on the fluorometer [???] to see if the complex could recognize a certain pollutant, and then carried out the full experiment. Therefore, the whole experiment is completely reliable” (in reply to the question about the absence of replicates).

I still believe that the manuscript can only be published after a major revision including new experiment.

Author Response

Dear Editor,

Here is our reply to the reviewer's question:

There are many research groups doing experiments on pollutant fluorescence recognition, and the whole experimental procedure is very mature. Please refer to the following four literatures [1-4], in which the whole experimental content was not repeated.

At the beginning of the experiment, we first conducted a preliminary fluorescence recognition test on the selected pollutants to see which kinds of pollutants can be recognized by the complex, and then we started to repeat the whole identification experiment. The experimental results are enough for our discussion.

Our research group has sufficient funds, and we know that molecules is a high-quality journal. We published several previous results on Molecules, and we will continue to consider publishing our best scientific research in the journals of mdpi in the future.

Maybe the following three reviewers are more favorable for the further review for the current manuscript:

lihuijunxgy@hpu.edu.cn; mengwei@ncst.edu.cn; libony0107@nynu.edu.cn

Sincerely,

Prof. & Dr. Cheng-Xing Cui

References

[1] Yajuan Mu, Yungen Ran, Bingbing Zhang, et al. Dicarboxylate Ligands Modulated Structural Diversity in the Construction of Cd(II) Coordination Polymers Built from NHeterocyclic Ligand: Synthesis, Structures, and Luminescent Sensing. Cryst. Growth Des. 2020, 20, 6030−6043.

[2] Yajuan Mu, Yungen Ran, Jianlong Du, et al. A fluorescent lanthanide-organic framework for highly sensitive detection of nitroaromatic explosives. Polyhedron 2017, 124, 125–130.

[3] Liangjuan Liu, Yungen Ran, Jianlong Du, et al. A luminescent Cd(II) coordination polymer as a multi-responsive fluorescent sensor for Zn2+, Fe3+ and Cr2O72- in water with fluorescence enhancement or quenching. RSC Adv., 2021, 11, 11266.

[4] Huijun Li, Yaling He, Qingqing Li, et al. Highly sensitive and selective fluorescent probe for Fe3+ and hazardous phenol compounds based on a water-stable Zn-based metal-organic framework in aqueous media. RSC Adv., 2017, 7, 50035.

Reviewer 2 Report

I am satisfied with the revisions the authors have made. 

Author Response

Dear Editor :

Thank you very much!

Sincerely yours,

Cheng-Xing Cui